# Endocannabinoid 2-Arachidonoylglycerol Synthesis and Metabolism at Neuronal Nuclear Matrix Fractions Derived from Adult Rat Brain Cortex

**DOI:** 10.3390/ijms24043165

**Published:** 2023-02-05

**Authors:** Xabier Aretxabala, Gontzal García del Caño, Sergio Barrondo, Maider López de Jesús, Imanol González-Burguera, Miquel Saumell-Esnaola, María Aranzazu Goicolea, Joan Sallés

**Affiliations:** 1Department of Neurosciences, Faculty of Pharmacy, University of the Basque Country (UPV/EHU), 01006 Vitoria-Gasteiz, Spain; 2Bioaraba, Neurofarmacología Celular y Molecular, 01008 Vitoria-Gasteiz, Spain; 3Department of Pharmacology, Faculty of Pharmacy, University of the Basque Country (UPV/EHU), 01006 Vitoria-Gasteiz, Spain; 4Centro de Investigación Biomédica en Red de Salud Mental (CIBERSAM), 28029 Madrid, Spain; 5Department of Analytical Chemistry, Faculty of Pharmacy, University of the Basque Country (UPV/EHU), 01006 Vitoria-Gasteiz, Spain

**Keywords:** 2-AG, neuronal nuclear matrix, DGLα, MGL, COX2, ABHD12

## Abstract

In this report, we describe the kinetics characteristics of the diacylglycerol lipase-α (DGLα) located at the nuclear matrix of nuclei derived from adult cortical neurons. Thus, using high-resolution fluorescence microscopy, classical biochemical subcellular fractionation, and Western blot techniques, we demonstrate that the DGLα enzyme is located in the matrix of neuronal nuclei. Furthermore, by quantifying the 2-arachidonoylglycerol (2-AG) level by liquid chromatography and mass spectrometry when 1-stearoyl-2-arachidonoyl-sn-glycerol (SAG) was exogenously added as substrate, we describe the presence of a mechanism for 2-AG production through DGLα dependent biosynthesis with an apparent *K_m_* (*K_m_*^app^) of 180 µM and a *V_max_* of 1.3 pmol min^−1^ µg^−1^ protein. We also examined the presence of enzymes with hydrolytic and oxygenase activities that are able to use 2-AG as substrate, and described the localization and compartmentalization of the major 2-AG degradation enzymes, namely monoacylglycerol lipase (MGL), fatty acid amide hydrolase (FAAH), α/β-hydrolase domain 12 protein (ABHD12) and cyclooxygenase-2 (COX2). Of these, only ABHD12 exhibited the same distribution with respect to chromatin, lamin B1, SC-35 and NeuN as that described for DGLα. When 2-AG was exogenously added, we observed the production of arachidonic acid (AA), which was prevented by inhibitors (but not specific MGL or ABHD6 inhibitors) of the ABHD family. Overall, our results expand knowledge about the subcellular distribution of neuronal DGLα, and provide biochemical and morphological evidence to ensure that 2-AG is produced in the neuronal nuclear matrix. Thus, this work paves the way for proposing a working hypothesis about the role of 2-AG produced in neuronal nuclei.

## 1. Introduction

Endocannabinoids (eCBs) play an important role in a diverse range of physiological brain processes during ontogenic neural development as adult postmitotic neural cells [1]. To date, two derivatives of arachidonic acid (AA), namely anandamide (AEA) and 2-araquidonoylglycerol (2-AG), identified nearly 30 years ago [2,3,4], are still considered the primary endogenous mediators of cannabinoid synaptic signaling through the activation of cannabinoid receptors. Other endogenous molecules generated from other fatty acid substrates, including palmitoylethanolamide [5] and oleoylethanolamide [6], have been identified that exert cannabinoid-like effects, although the evidence for interaction with cannabinoid receptors does not yet appear to be definitive. For nearly two decades, whole brain AEA and 2-AG content has led to debate regarding the physiological range of eCB concentrations in various cerebral regions. In fact, while analysis of whole rodent brain estimates 2-AG content to be approximately 1000-fold higher than AEA content (nmol g^−1^ vs. pmol g^−1^, respectively) [7], in vivo microdialysis of interstitial eCBs from awake behaving animals that provide an index of “signaling competent” eCB levels [7,8], estimates 2-AG levels to be less than 10-fold higher than AEA in several brain structures [9,10,11]. One mechanism that may help explain why the 2-AG/AEA ratio measured in whole brain tissue is much higher than in in vivo microdialysates is the rapid increase in 2-AG content that occurs postmortem [7,12]. However, a relative proportion of the 2-AG content of brain tissue is localized to intracellular sites. At present, it is well established that the diacylglycerol (DAG) coming from phospholipase C β1 (PLCβ1) activity is taken by diacylglycerol lipase-α (DGLα) to produce 2-AG, which is the most abundant endogenous agonist of the CB1 cannabinoid receptor [13,14,15,16]. The postsynaptic localization of the molecular architecture of this “on-demand” signaling cascade has been accurately defined [17,18,19,20,21], where PLCβ1 acts as a coincidence detector that integrates signals from Gq-coupled receptors with extracellular calcium entry via ionotropic receptors or voltage-gated calcium channels [13,14,15,16,22,23,24]. In the brain, the hydrolysis of 2-AG to AA is mainly mediated by monoacylglycerol lipase (MGL), while the hydrolysis of 2-AG to AA catalyzed by α/β-hydrolase domain 6 (ABHD6) and 12 (ABHD12) proteins account for 15% of the total enzymatic conversion [25]. In some conditions, 2-AG is also susceptible to being oxidized by cyclooxygenase-2 (COX2) into the prostaglandin H_2_ (PGH_2_) glycerol ester PGH_2_-G [26]. Finally, fatty acid amide hydrolase (FAAH), the primary enzyme catalyzing the hydrolysis of anandamide, has also been reported to be involved in the physiological inactivation of 2-AG [27,28].

In spite that most of the research on signal transduction pathways based on PLCβ1/DGLα has been devoted to the study of phenomena that take place at the plasma membrane, the work of several independent laboratories has consistently demonstrated that the phosphoinositide cycle (biosynthetic and hydrolytic machinery) is present in the cell nucleus [29,30]. We have previously demonstrated in neuronal nuclei isolated from adult rat brain cortex that PLCβ1 is localized to chromatin-poor domains of the nuclear matrix, where it displays a high co-localization with NeuN/Fox-3 and SC-35 [31], two known markers of nuclear speckles [32,33,34]. More recently, we explored whether a PLCβ1/DGLα-dependent machinery for 2-AG synthesis was actually present in the nuclear compartment of rat brain cortical neurons [35]. Our results obtained with highly-resolution fluorescence microscopy in histological sections of adult rat brain cortex and purified nuclei make it plausible that a DGLα-dependent synthesis of 2-AG exists in the neuronal nucleus.

With this evidence in mind, we focused here on the kinetics of DGL activity leading to the generation of 2-AG within the nuclear matrix subcompartment of neuronal nuclei isolated from the cerebral cortex of the adult rat brain, and on the study of 2-AG metabolizing brain enzymes as potential candidates to regulate 2-AG levels and signaling mediated by this eCB in the nuclear matrix.

## 2. Results

### 2.1. Analysis of Diacylglycerol Lipase Enzymatic Activity and Pharmacological Assessment of Serine Hydrolase-Dependent Degradation of 2-AG in Nuclear Matrix Subfractions

Because the two enzymes responsible for endogenous 2-AG production, i.e., PLCβ1 and DGLα, are co-localized in the discrete nuclear matrix (NM)-associated domains of neuronal nuclei isolated from the adult rat cerebral cortex [31], we chose this subfraction to measure DGL enzymatic activity, as well as to assess the role of serine hydrolase activities in 2-AG degradation.

#### 2.1.1. Measurement of Diacylglycerol Lipase Enzymatic Activity

In all experiments, we used the DAG species 1-stearoyl-2-arachidonyl-sn-glycerol (SAG) as a substrate for DGL enzymatic activity and measured accumulated 2-AG by liquid chromatography–tandem mass spectrometry (LC/MS/MS). For initial experiments aimed at establishing the optimal experimental conditions for the measurement of DGL activity, the SAG concentration was set at 200 µM, which is close to the *K_m_* value of 158 µM obtained by Shonesy et al. in cell membranes [36]. Under these conditions, analysis of 2-AG production as the amount of NM protein increased showed a linear trend between 5 and 10 µg total protein (Figure 1A), while 2-AG accumulation over time in a fixed amount of 10 µg of nuclear protein showed linearity between 5 and 45 min (Figure 1B). Furthermore, DGL activity increased with the addition of the reducing agent dithiothreitol (DTT) in a concentration-dependent manner within the range of 2 to 10 mM DTT, with 10 mM being the optimal concentration. (Figure 1C). When we examined the effect of Triton X-100 (0.1%, v/v) on the concentration-dependent effects of DTT, we observed that this non-ionic detergent significantly inhibited 2-AG production (Figure 1C). Under these optimal reaction conditions (10 µg total protein, 15 min, 10 mM DTT), the addition of 10 µM tetrahydrolipstatin (THL), an inhibitor of DGL activity, completely abolished 2-AG production from 200 µM SAG substrate, revealing that 2-AG accumulation was dependent on DGL activity (Figure 1D). Although in these preliminary assays we did not detect any accumulation of arachidonic acid (AA), as a consequence of possible hydrolytic activities in the NM samples, we performed additional assays in the presence of selective inhibitors of the enzymes with serine hydrolase activity MGL and ABHD6. Neither the MGL inhibitors NAM (10 µM) and JZL184 (10 µM) nor the ABHD6 inhibitor WWL70 (10 µM) had any effect on 2-AG production (Figure 1D), thus ruling out the possibility that these activities affect the accumulation of 2-AG in our experimental conditions. On the other hand, to rule out possible effects of a cyclooxygenase activity on the accumulation of 2-AG, we performed assays in the presence of 10 µM R-flurbiprofen, which was chosen based on its properties as a selective inhibitor of COX2 according to previous reports [25,37,38,39,40]. Therefore, since the 2-AG generated as a result of DGL activity was not subsequently degraded to AA, we conclude that the experimental conditions established for the measurement of 2-AG accumulation in nuclear matrix samples are adequate to study the enzymatic kinetics of DGL activity.

Once the optimal experimental conditions for the measurement of DGL activity-dependent 2-AG accumulation were established, we performed assays using seven concentrations of the SAG substrate within a range of 30–500 µM, and fitted the 2-AG generation progress curves to the Michaelis–Menten equation (Figure 2). Thus, we calculated the Michaelis–Menten constant (*K_m_*) and the maximum rate of the enzymatic reaction (*V_max_*) for this substrate, obtaining values of *K_m_*^app^ = 179.8 ± 15.8 µM and *V_max_* = 1.3 ± 0.22 pmol min^−1^ µg^−1^ protein (mean ± SEM). The calculated *K_m_*^app^ value is consistent with that previously found in membranes of DGLα-transfected HEK293T cells [2], suggesting that the DGLα isoenzyme is responsible for 2-AG production in our sample. Instead, the *V_max_* value obtained here was significantly lower than that observed by Shonesy et al. [36] in the membranes isolated from cells that overexpresses DGLα (1.3 ± 0.22 vs. 9.8 ± 1.1 pmol min^−1^ µg^−1^ protein).

#### 2.1.2. Serine Hydrolase Activity-Dependent 2-AG Degradation in the Nuclear Matrix

Our results showing no detectable levels of AA in NM samples after the addition of exogenous SAG, along with the fact that selective inhibitors of serine hydrolase activities had no effect on the accumulation of 2-AG (Figure 1), indicate that the 2-AG produced by DGL activity is not attacked by endogenous serine hydrolases in our experimental conditions. However, rather than the lack of available enzymes with monoacylglycerol lipase activity (MGL/ABHD6/ABHD12), this observation could be due to the fact that 2-AG concentrations of around 2 µM achieved in our optimized assay are too low to “turn on” the activity of serine hydrolase enzymes that might be present in NM samples. To test this possibility, we first examined whether exogenous 2-AG added to the NM samples at a higher concentration could be degraded to AA. Thus, we added 2-AG (10 µM) or its isomer 1-AG (10 µM) as exogenous substrates, and we measured AA production by LC/MS/MS. The experiments were performed under the same experimental conditions used to measure DGL activity, except that the incubation time was extended to 30 min. The results showed that the addition of either of the two monoacylglycerol isomers (2-AG or 1-AG) to NM samples at a concentration of 10 µM resulted in AA production that accounted for about 20% of the added substrate (Figure 3).

### 2.2. Distribution of 2-AG Metabolizing Enzymes in Subcellular and Subnuclear Fractions

#### 2.2.1. Analysis of the Enrichment of Cortical Intact Nuclei in 2-AG Metabolizing Enzymes

As described above, the kinetics of 2-AG production from exogenously added SAG in NM samples is consistent with a major role for the DGLα isoenzyme in this activity. Although our data clearly showed that 2-AG accumulation in these assays was not affected by the activity of 2-AG metabolizing enzymes with serine hydrolase or cyclooxygenase activities, the presence of serine hydrolase activity was clearly confirmed when exogenous 2-AG was added, raising the possibility that this activity might be physiologically relevant. With this idea in mind, we analyzed the enrichment of cortical intact nuclei in 2-AG metabolizing enzymes as the likely candidates responsible for AA production in nuclear matrix samples. To this end, the expression of fatty acid amide hydrolase (FAAH), monoacylglycerol lipase (MGL) and α/β hydrolase domain-containing protein 12 (ABHD12) was analyzed by immunoblot in whole homogenates (WH) of the adult rat brain cortex, and intact nuclear (N), crude membrane (P2), cytosolic (S3) and microsomal (P3) fractions. In addition, the expression of COX2, another 2-AG metabolizing enzyme, was analyzed in the same fractions.

To check the quality of the subcellular fractionation procedure, equal amounts of proteins from WH and N, P2, P3 and S3 fractions were resolved side by side on the same gel by SDS-PAGE and analyzed by Coomassie-blue staining and immunoblot with antibodies against different subcellular structures. Proteins from WH and N, P2, S3 and P3 fractions resolved by SDS-PAGE produced clearly different Coomassie-stained polypeptide profiles (Figure 4A). Of particular interest for the current study was the enrichment in polypeptides corresponding to histones in the N fraction, whose purity was also checked by phase contrast microscopy (Figure 4B). Total protein in cortical homogenates preferentially partitioned to the cytosol (3.22 ± 0.30 mg/100 mg tissue) and crude membranes (2.58 ± 0.11 mg/100 mg tissue), followed by microsomes (0.72 ± 0.15 mg/100 mg tissue) and intact nuclei (0.08 ± 0.02 mg/100 mg tissue) (Figure 4C). Western blot analysis of subcellular fractions using specific markers for nuclear, plasma membrane, endoplasmic reticular and cytosolic fractions confirmed the suitability of the fractionation method (Figure 4D). Thus, intense immunoreactive bands for nuclear markers Histone H3 and NeuN/Fox-3 were detected in N fraction, and no immunoreactivity could be detected in crude membranes (P2), microsomes (P3) and cytosol (S3). By contrast, immunoreactive bands for the plasma membrane specific markers α1 subunit of the Na^+^/K^+^ ATPase and thymocyte antigen 1 (Thy-1) were intense in P2 and P3 membrane pellets, but provided no signal in N and S3 fractions. Antibodies against the binding immunoglobulin protein (BiP), a chaperone located in the lumen of the endoplasmic reticulum (ER), yielded an intense immunoreactive band in P3 and, to a lesser extent in N fraction, consistent with the enrichment of P3 in ER membranes and with the presence of this protein in the perinuclear space of the nuclear envelope, a region contiguous with the lumen of the endoplasmic reticulum. Intense BiP-immunoreactivity was also seen in P2 fraction, which probably results from partial contamination of P2 with ER membranes. Weak BiP immunoreactivity in S3 was probably due to the release of soluble proteins (including BiP) from ER lumen during homogenization. Finally, immunoreactivity for the cytosolic marker β-tubulin was very intense in the S3 fraction, although faint signals could also be observed in membrane pellets P2 and P3 (Figure 4D).

Western blot analysis using antibodies to MGL, FAAH, ABHD12 and COX2 yielded major bands in WH with apparent molecular masses roughly consistent with those expected: 35.5, 63.6, 41.4–45.1 and 69.2 kDa, respectively (Figure 5A). The anti-ABDH12 goat polyclonal antibody (against the 13 amino acids of the extreme C-terminus common to ABDH12 isoforms 1 and 3) produced two intense immunoreactive bands, with the upper one migrating slightly above the 50 kDa standard, at a size considerably larger than the theoretically predicted mass of the ABHD12 isoform 1 (45.1 kDa). Because several antibodies from a different host and raised against a different epitope produce a similar double band pattern (e.g., rabbit monoclonal antibodies ab182011 and ab250575 from Abcam, Cambridge, UK), both bands were considered antigen-specific. Finally, COX2 yielded a cluster of close bands consistent with the presence of multiple glycosylation sites. Noticeably, we found intense immunoreactivity for MGL-, ABHD12- and COX2 proteins in samples of intact nuclei, although with some differences in the migration patterns of ABHD12 and COX2 with respect to WH. Thus, ABHD12-immunoreactive bands were detected at about 5 kDa above those in WH and P2, suggesting that post-translationally modified forms are present in the nucleus. In its turn, COX2 consisted of a single very intense band, which contrasted with the multi-band pattern observed in WH and P2. Semi-quantitative analysis of the subcellular distribution of MGL, FAAH, ABHD12 and COX2 was performed by densitometric analysis of immunoreactive bands. The optical density of immunoreactive bands corresponding to incremental amounts of protein from P2 fraction was measured and standard curves were generated by linear regression analysis. For all four enzymes, analysis of the standard curves revealed a linear relationship between the amount of P2 protein and the relative optical density and, therefore, the slopes obtained could be used to calculate the relative expression in N, P2, P3 and S3 fractions, which was normalized to that in the crude lysate (WH) (Figure 5B). As a result of this analysis, we concluded that MGL, ABHD12 and COX2 are expressed in intact nuclei. Moreover, MGL and, to a lesser extent, ABHD12 were highly enriched in intact nuclei with respect to the crude cortical lysate (Figure 5B).

On the basis of results from Western blot experiments showing that MGL, ABHD12 and COX2 (but not FAAH) are expressed in intact nuclei from the adult rat brain cortex, double-immunofluorescence experiments were conducted to analyze if these enzymes are expressed in neuronal and/or glial nuclei. Thus, polyclonal antibodies to MGL, ABHD12 and COX2 were combined for double immunofluorescence in intact nuclei with a monoclonal antibody against NeuN/Fox-3, widely used as a marker of postmitotic neurons. Intense immunoreactivity was observed for the three enzymes in medium to large-sized NeuN/Fox-3 positive nuclei. MGL- and ABHD12-staining was absent, or barely detectable above the background, in non-neuronal NeuN/Fox-3 negative nuclei, whereas clear COX2-immunoreactivity could be observed in most of these nuclei (Figure 6).

#### 2.2.2. Subnuclear Partitioning of 2-AG Metabolizing Enzymes

The distribution of MGL, ABHD12 and COX2 within the nucleus and with respect to DGLα was then analyzed by double immunofluorescence and counter-staining with Hoechst’s chromatin staining, followed by high-resolution fluorescence microscopy. For this purpose, antibodies against MGL, ABHD12 and COX2 were combined with antibodies against the component Nup62 of the nuclear pore complex (NPCx), the intrinsic component of the neuronal nuclear matrix NeuN/Fox-3 and DGLα (Figure 7, Figure 8 and Figure 9). The subnuclear distribution and staining pattern of MGL and ABHD12 were roughly similar. Both signals were enriched in bright foci distributed throughout the nucleoplasm, in subdomains displaying weak Hoechst’s chromatin staining. Double immunolabeling with NPCx clearly showed that immunoreactivity corresponding to MGL and ABDH12 was internal to the nuclear envelope, whereas there was considerable but not complete overlap with the signals corresponding to the nuclear matrix marker NeuN/Fox-3 and to DGLα (Figure 7 and Figure 8).

In contrast with the distribution of MGL and ABHD12 signals, immunostaining with an antibody to COX2 produced a ring pattern around the nucleus, which was more intense in the largest neuronal nuclei compared to small and medium-sized ones. Additionally, a weaker and more diffuse signal was evident in nucleoplasm regions with poor chromatin staining. Consequently, a high co-localization was observed between COX2 and NPCx, whereas only a few bright COX2-positive spots exhibited overlap with NeuN/Fox-3 and DGLα (Figure 9).

To further analyze the subnuclear compartmentalization of MGL, ABHD12 and COX2 enzymes, intact nuclei (N) isolated from homogenates of adult rat cortex were subfractioned by a two-step sequential treatment with the non-ionic detergent TX-100, followed by DNase I/high salt extraction. Thus, two nuclear pellets and two supernatants were obtained in each step. After treatment with Triton X-100, the first pellet of nucleoids (Nu) consists of intact nuclei depleted of nuclear envelope and nucleoplasm soluble proteins, which are recovered in the S1 supernatant. Treatment of nucleoids with DNase I followed by high salt results in a second pellet, mostly referred to as the nuclear matrix (NM), a remaining structural framework consisting of a peripheral layer of pore-complexes in association with a lamina, residual nucleoli and internal fibrillar structures [41,42]. The NM fraction is enriched in RNA-bound and depleted of DNA-bound proteins, which are recovered in the S2 supernatant. Of the total protein recovered after treatment with Triton X-100, 87.8 ± 3.9% and 12.2 ± 3.1% (mean ± SD) was found in the Nu and S1 fractions, respectively. Protein recovered in the NM and S2 fractions represented 40.3 ± 3.9% and 47.5 ± 7.1% of the total nuclear protein, respectively (Figure 10A). Treatment of intact nuclei with DNase I and high salt resulted in a marked decrease in the Coomassie Brilliant Blue staining of protein bands corresponding to linker (32–34 kDa) and core histones (15–17 kDa) in the NM with respect to Nu fraction (Figure 10B). Accordingly, histone staining was highest in the S2 and absent in the S1 supernatants. Immunoblot analysis of nuclear subfractions using an antibody against BiP demonstrated that the nuclear envelope and nucleoplasm soluble proteins were efficiently solubilized by Triton X-100, as BiP-immunoreactivity was nearly undetectable in nucleoids and recovered in the Triton X-100 soluble supernatant (Figure 10C). Consistent with Triton X-100 and DNase I-high salt resistance of Nup62, NPCx-immunoreactivity was clearly detectable in Nu and considerably increased in NM due to the removal of the large pool of DNA-bound proteins from this fraction. This finding was also indicative that structural integrity was preserved during fractioning. Similarly, sequential detergent and DNase I/high salt treatments could not deplete NeuN/Fox-3 immunoreactivity, which increased in NM fraction, in agreement with the fact that a portion of NeuN/Fox-3 is associated with the internal ribonucleoprotein network [34]. As expected, the DNA-associated Histone H3 protein showed an opposite behavior to that observed for NeuN/Fox-3. (Figure 10C). Once the effectiveness of the fractioning protocol was demonstrated, the nuclear compartmentalization of MGL, ABHD12 and COX2 enzymes was analyzed by Western blot analysis. Results indicate that nuclear MGL and COX2 are mainly DNA-bound and non-ionic detergent soluble proteins, respectively (Figure 10D). In regards to ABHD12, the signal corresponding to the upper and lower bands found in intact nuclei behaved differently. Thus, the upper one was almost released from nucleoids by DNase I/high salt, whereas the lower one was partially sensible to detergent extraction but resistant to DNase I/high salt treatment. Moreover, immunoreactivity for the lower ABHD12 band was only slightly decreased by Triton X-100 treatment and considerably increased in NM following the solubilization of DNA-bound proteins (Figure 10D). Of note, because the same amount of Triton X-100 insoluble (in Nu fraction) and soluble proteins (in S1 fraction) was loaded for immunoblot analysis (whereas ~9.6-fold more protein was recovered in the Nu pellet), our results reveal that only a small portion of the lower ABHD12-immunoreactive band was non-ionic detergent sensitive. Taken together, these results point to ABDH12 enzyme as a possible candidate for being responsible for the 2-AG hydrolytic activity observed in NM samples.

### 2.3. Pharmacological Characterization of the ABDH12-Dependent Hydrolysis of 2-AG in Nuclear Matrix Subfractions

Taken together, the data described so far point to ABDH12 as the enzyme responsible for the observed hydrolase activity leading to AA accumulation in the NM samples of intact adult rat cortical nuclei. Indeed, AA production in NM samples was not sensitive to selective MGL and ABHD6 inhibitors, whereas ABDH12 was the only 2-AG hydrolyzing enzyme detected in the nuclear matrix as assessed by Western blot on nuclear subfractions and double immunofluorescence in intact nuclei. Thus, we explored the possibility that ABHD12 enzyme could be responsible for the observed hydrolysis of 2-AG. To this end, in the absence of a selective inhibitor, we tested the ability of methyl-arachidonoyl fluorophosphonate (MAFP) (a potent irreversible inhibitor of ABHD12 activity but also of ABHD6 and FAAH activities [40,43,44]) to block AA production from exogenously added 2-AG. In our hands, MAFP was very effective in reversing AA production in a dose-dependent manner. Thus, the AA accumulated in 30 min from exogenous 10 µM 2-AG decreased significantly as MAFP concentration increased (AA concentration values at baseline and after adding 30 nM MAFP were 2.39 ± 0.01 µM and 0.43 ± 0.04 µM, respectively). Inhibition showed a clear concentration dependence with a calculated inhibitory potency (*IC_50_* = 1 nM) (Figure 11) consistent with the characteristics of the interaction between MAFP and ABHD12. In conclusion, the experimental evidence shown here indicates that ABHD12 is the main responsible for the hydrolysis of 2-AG in the nuclear matrix (Figure 12).

## 3. Discussion

In previous research, we examined the subcellular distribution of the endocannabinoid 2-AG production machinery in the adult rat cerebral cortex, revealing that the enzymes involved are also expressed in the nucleus of adult neurons. Thus, using high-resolution fluorescence microscopy, subcellular fractionation, and Western blot techniques, we demonstrated that the DGLα enzyme was located in the matrix of neuronal nuclei. Furthermore, we unequivocally demonstrated by LC/MS/MS that a PLCβ1/DGLα signaling cascade leading to endogenous 2-AG production operates in whole nuclei isolated from rat cerebral cortex [31,35,45]. With this background, here we used nuclear matrix samples extracted from intact nuclei to estimate by LC/MS/MS the kinetic parameters of DGL activity-dependent 2-AG accumulation using the DAG species SAG as an exogenous substrate. We also examined the presence of the main enzymatic activities responsible for the degradation of 2-AG and analyzed, by double immunofluorescence and high-resolution fluorescence microscopy and by biochemical subcellular fractionation and Western blot, the expression in neuronal nuclei and the subnuclear compartmentalization of the main 2-AG degrading enzymes; namely, MAGL, FAAH, ABHD12, and COX2. Of these, only ABHD12 was found to be localized to the nuclear matrix and, furthermore, inhibitors of the ABHD family of serine hydrolases, but not MAGL inhibitors, prevented 2-AG hydrolysis. Overall, our results provide new insights into the subcellular distribution of neuronal DGLα and add demonstrative biochemical and morphological evidence that 2-AG is produced in the nuclear matrix of neuronal nuclei, thus paving the way to propose working hypotheses on the pathophysiological role of 2-AG produced in neuronal nuclei.

Using SAG as a substrate added exogenously to nuclear matrix fractions, we obtained an estimated value of the Michaelis–Menten constant (*K_m_*^app^) for DGL activity close to that observed in membranes of DGLα-transfected cells [36]. By contrast, the value of *V_max_* obtained here was about one-quarter with respect to the one observed in the mentioned cellular system. This difference is most likely due to a very high expression level of DGLα in the membranes of the transfected cells compared to the endogenous levels in our samples. However, the duration and magnitude of 2-AG signaling is thought to depend on the balance between production and degradation, and the most studied pathway that controls this balance is the breakdown of 2-AG into AA and glycerol [46] by serine hydrolases [25,38,47,48,49,50,51]. Although most studies show that the major brain 2-AG degrading enzyme is MGL [25,49,52], 2-AG produced in nuclei from endogenous or exogenously added substrates appears not to undergo MGL-catalyzed hydrolysis. Indeed, the omission of MGL inhibitors had little effect on endogenous 2-AG production driven by activation of the PLCβ1/DGLα nuclear signaling axis [31] and, in the same line of evidence, when we here measured the accumulation of 2-AG from the DAG species SAG exogenously added to the nuclear matrix samples (DGL activity), we found no traces of AA formation. Furthermore, neither the MGL inhibitors NAM or JZL184 nor the ABHD6 inhibitor WWL70 favored 2-AG accumulation from exogenous SAG. However, 20% of exogenously added 10 µM 2-AG was hydrolyzed to AA, demonstrating the presence of serine hydrolase activity in the NM samples. The lack of in-concert DGL and serine hydrolase activities leading to AA production under our optimized experimental conditions for the measurement of DGL activity-dependent 2-AG accumulation could be due to the unavailability of the 2-AG formed for hydrolysis catalyzed by serine hydrolases or, alternatively, to the inability of the low 2-AG concentrations achieved in these assays (around 2 µM) to activate enzymes with serine hydrolase activity (MGL, ABHD6, ABHD12). For example, the *K_m_* of 2-AG for the ABHD12 enzyme is greater than 100 µM [40]. Apart from MGL, three other serine hydrolases capable of degrading 2-AG are expressed in the rodent brain [25,49,52], all of which are located in the synapse [53]. The most detailed study of the contribution of these serine hydrolases to the degradation of brain 2-AG was carried out by Blankmann et al. [25]. As observed by these authors in mice brain membranes, the contribution of MGL, ABHD12, and ABHD6 enzymes to the hydrolysis of 2-AG is 85, 9, and 4%, respectively. Among all the other enzymes studied, FAAH was the next most active hydrolase, and although its main substrate in vivo is anandamide (AEA), FAAH is effective as a 2-AG hydrolase in in vitro assays. [28,54]. Despite the contribution of FAAH only represents about 1% of the total 2-AG degradation activity in mice brain membranes [25], the relative expression levels of these enzymes could be very different in two structurally and functionally distant cellular domains such as the synapse and the nucleus. Therefore, under the premise that some of these four serine hydrolases might be located in the nucleus (based on our experimental evidence showing 2-AG degradation activity in the nuclear matrix), we examined by immunoblot the partitioning of MGL, FAAH, and ABHD12 enzymes during subcellular fractionation of whole homogenates (WH) from adult rat cerebral cortex into the plasma membrane (P2), intact nuclear (N), cytosolic (S3), and microsomal (P3) fractions. We were unable to analyze the expression of ABHD6 as we could not find a commercially available antibody against this enzyme. Noteworthy, the immunoreactive signals consistent with the theoretical molecular mass of MGL and ABHD12 were more intense in the N fraction than in crude homogenates for equal amounts of protein, indicating that the nuclei of the adult rat cerebral cortex are enriched in these two hydrolases. Moreover, MGL immunoreactivity was approximately three times more intense in the nuclear (N) fraction than in the plasma membrane (P2) fraction, while ABHD12 showed similar immunoreactivity in both subfractions. In a similar way to what has been described in the synapse, where the expression level of serine hydrolases in different subdomains explains, at least in part, their contribution to the degradation of 2-AG [53], the study of the partitioning of these enzymes in the different nuclear subdomains could shed light on their contribution to the 2-AG degrading activity observed in the nuclear matrix. Our double immunofluorescence assays on intact nuclei showed that immunoreactive signals for both MGL and ABHD12 were strong at discrete areas of the nucleoplasm with weak chromatin staining and showed high but incomplete overlap with both NeuN/Fox-3 (nuclear matrix marker) and DGLα signals. Consistent with these observations, Western blot assays on equal amounts of protein from nuclear subfractions showed that immunoreactivity for both MGL and ABHD12 was maintained and even increased in the nucleoid (Nu) fraction after nuclear envelope solubilization by non-ionic detergent treatment. By contrast, MGL-immunoreactivity was virtually absent in the nuclear matrix (NM) fraction after chromatin digestion with DNase I, while a large pool of the ABHD12 signal remained in the nuclear matrix. Noteworthy, the two bands produced by the ABHD12 antibody behaved differently, the upper one being partially sensitive to DNase I extraction but not TX-100 and the lower one to TX-100 but not DNase I extraction, suggesting the presence of two ABHD12 variants as part of different intranuclear signaling complexes. In any case, of the three enzymes analyzed, both MGL and ABHD12 (but not FAAH) were present in isolated nuclei, although only ABHD12 was associated with the nuclear matrix, a nuclear subdomain in which DGLα is also present, as we have shown previously [35]. Taken together, these results indicate that ABHD12 is the major hydrolytic enzyme for 2-AG production in the nucleus and are consistent with the fact that the omission of MGL inhibitors does not affect either endogenous 2-AG production in intact nuclei [35] or 2-AG production from exogenous substrates here observed in the nuclear matrix. To further confirm this conclusion, we analyzed the capacity of the serine hydrolase inhibitor MAFP to reverse the conversion of exogenous 2-AG to AA, showing that it completely inhibited AA accumulation in a concentration-dependent manner with a *IC_50_* around 1 nM, which is consistent with the affinity of MAFP for the ABHD12 enzyme [40]. Although the MAFP compound used in this assay is not a selective inhibitor of ABHD12, but also of ABHD6 and FAAH [40,43,44], the set of results lead to the conclusion that the inhibitory effect on 2-AG hydrolysis observed in this assay involves only ABHD12 and is, therefore, the enzyme responsible for the catalytic activity observed in the nuclear matrix (Figure 12). First, AA was undetectable by LC/MS/MS in our optimized conditions for the measurement of 2-AG production from exogenous SAG in the NM samples. Second, neither MGL nor ABHD6 inhibitors favored 2-AG accumulation from exogenous SAG, while their omission did not affect 2-AG production. Third, of the serine hydrolases analyzed in subfractions of intact nuclei isolated from the adult rat cortex, ABHD12 was the only one present in the nuclear matrix. Fourth, MAFP inhibited 2-AG hydrolysis in a concentration-dependent manner with a *IC_50_* consistent with the affinity of MAFP for the ABHD12. However, despite the role that ABHD12 appears to play in the hydrolysis of 2-AG in the nuclear matrix, our results do not exclude the possibility that other serine hydrolases may degrade 2-AG in other nuclear subdomains. Indeed, taken together, results of double immunofluorescence and Western blot analyses lead to the conclusion that MGL localizes to DNase I-sensitive foci of the nucleoplasm, where it could be involved in 2-AG degradation. Therefore, it is tempting to speculate that the heterogeneous distribution of the serine hydrolases MAGL and ABHD12 in the nucleus might allow control of different pools of 2-AG, as has been proposed for 2-AG degradation at the plasma membrane [25]. As addressed in mouse brain membranes, an activity-based protein profiling (ABPP) approach [25] in combination with nonselective inhibitors and serine hydrolases could shed light on the contribution of different molecular entities modulating 2-AG levels in different nuclear compartments. 

Another metabolic pathway potentially contributing to the regulation of 2-AG levels is COX2-dependent oxidation. Thus, several studies have shown that COX2 activity has a significant effect on the function of 2-AG in retrograde synaptic signaling. For example, inhibition of COX2 activity in hippocampal slices increases the duration of DSI [55], suggesting that COX2 may be involved in limiting retrograde 2-AG signaling. Similarly, inhibition of COX2 decreases excitatory transmission in the CA1 hippocampal region by a CB1 receptor-mediated mechanism [56] and MAGL and COX2 cooperate in adjusting DSI in a subpopulation of hippocampal inhibitory neurons [57]. In any case, it is clear that COX2 activity affects the accumulation of 2-AG in the synapse and could have a similar role in the nucleus. Both double immunofluorescence on isolated nuclei followed by high-resolution fluorescence microscopy and Western blot analysis on nuclear subfractions clearly showed that nuclear COX2 is targeted to the envelope, making it unlikely that COX2 activity could have any impact in the accumulation of 2-AG formed from exogenously added SAG in nuclear matrix samples. Indeed, the addition of the COX2 inhibitor R-flurbiprofen did not favor 2-AG accumulation in our experimental conditions for the measurement of DGL activity in nuclear matrix samples. However, as discussed above for the MGL and ABHD12 enzymes with serine hydrolase activity, it cannot be ruled out that COX2-dependent degradation of 2-AG could take place in whole nuclei. In this regard, during the last decades, the COX-mediated production of prostaglandin glycerol esters (PG-Gs) from endocannabinoids has attracted the attention of basic researchers [58]. Thus, it is well known that COX2 can convert AA into the cyclic endoperoxide prostaglandin G2 (PGG_2_) and subsequently reduce the unstable intermediate PGH_2_. Specific isomerases convert PGH_2_ into a variety of eicosanoids with a broad spectrum of biological effects including prostaglandins D2 (PGD_2_), E2 (PGE_2_), F2α (PGF_2α_) and I2 (PGI_2_ or prostacyclin) and thromboxane A2 (TXA_2_) [59]. Similarly, COX2 can metabolize 2-AG to the PG-Gs PGG_2_-G and PGH_2_-G, which are subsequently converted to PG-G analogous to the canonical prostaglandins (PGD_2_-G, PGE_2_-G, PGF_2α_-G and PGI_2_-G) with the exception of TXA_2_, since thromboxane synthase does not accept glycerol ester derivatives of PGH_2_ [58,60,61,62,63]. Of note, PG-Gs formed from 2-AG by COX2 activity exhibit in many systems potencies similar to that of AA-derived prostaglandins formed through the canonical pathway [58].

From the data set shown here, it can be concluded that the neuronal nucleus is endowed with a complex and compartmentalized machinery for the production and degradation of 2-AG, which in the nuclear matrix depends, respectively, on the activities of DGLα and AHBD12 enzymes, whereas MGL and COX2 are potential candidates for 2-AG hydrolysis and conversion to PG-Gs in the non-matrix nucleoplasm and nuclear envelope subdomains, respectively. Although the role of 2-AG produced in the nucleus will require further investigation, the experimental evidence shown here paves the way for generating plausible hypotheses that can be addressed in future studies. Among them, locally produced 2-AG could have a function as an endogenous ligand of the nuclear peroxisome proliferator-activated receptor γ (PPARγ) [64,65], a nuclear receptor with transcription factor activity that regulates multiple genes involved in a plethora of cellular processes including metabolism, proliferation, differentiation and survival. Interestingly, like PGD_2_, its glycerol ester derivative PGD_2_-G is readily dehydrated to produce the cyclopentenone-type prostaglandin 15-deoxy-Δ^12,14^-PGJ_2_ glycerol ester (15d-PGJ_2_-G), which is a PPARγ agonist as potent as 15d-PGJ_2_ [66,67]. Since PPARγ receptor agonists [68,69,70], including cannabinoid derivatives [71,72], favor neuron survival in different paradigms of neuroinflammation, it is tempting to speculate that 2-AG might act as a protector of neurons through the activation of the PPARγ receptor in situations of compromise of cellular homeostasis. Therefore, it would be interesting to study whether nuclear 2-AG accumulates under neuroinflammation or other conditions that compromise neuronal survival. In this sense, under conditions of hypoxia [73] and hyperexcitability [74], nuclear DAG kinase ζ translocates to the cytoplasm and protects neurons from p53-dependent death [74]. Of note, in our assays of 2-AG accumulation in whole nuclei isolated from the cortex of the adult rat [35], we found that omission of inhibitors of DAG kinase activity led to a significant reduction of 2-AG accumulation driven by in-concert PLCβ1 and DGLα activities, probably due to decreased DAG availability. It is likely that the translocation of nuclear DAG kinase ζ to the cytosol increases the availability of DAG in the nucleus for DGLα-dependent 2-AG production. Therefore, the study of the production and degradation of 2-AG in isolated nuclei of animals that have suffered hypoxia and hyperactivity would be very interesting. Another potential target of 2-AG belonging to the same family of transcription factors as PPARγ is the related nuclear receptor PPARα [75], whose activation is known to have anti-inflammatory effects [76]. Interestingly, the pharmacological blockade of endocannabinoid-inactivating hydrolases reverses PPARα-mediated anti-nociceptive effects [77]. In this regard, it has recently been proposed that AEA exerts part of its anti-nonciceptive effects through PPARα [78]. In view of our results, locally produced 2-AG in the nuclear matrix and its hydrolyzing enzymes could be equally candidates for a role in PPARα-dependent molecular mechanisms of pain. Although these hypotheses are still very premature and require further investigation, our results and the body of evidence in the literature suggest that the noncanonical pathway for 2-AG production in the neuronal nucleus may have relevant pathophysiological implications. Therefore, the importance of this signaling pathway for drug development deserves attention. Although clinical research has focused on potent inhibitors of FAAH and MAGL, with drugs even being licensed for clinical trials or neuroinflammatory and neurological research [79,80], the development of compounds that selectively inhibit enzymes of the ABHD family, including ABHD12, is still in its infancy. Our results, which show that ABHD12 is probably the most important serine hydrolase that tunes 2-AG production in the nuclear matrix, point to this enzyme as a potential drug target. Furthermore, ABHD12 inhibitors should be evaluated with this framework in mind.

## 4. Materials and Methods

### 4.1. Drugs, Chemicals and Antibodies

For the endocannabinoid determinations, 2-AG and 1-AG (and their deuterated analogs 2-AG-d5 and 1-AG-d5), AA (and its deuterated analog AA-d8) and SAG were obtained from Cayman Chemicals (Vitro SA, Madrid, Spain). Water (H_2_O), acetonitrile, formic acid, ethyl acetate, hexane and methanol (Fluka, LC/MS grade) were obtained from Sigma-Aldrich (Madrid, Spain). All stock solutions, intermediate dilutions and calibration standards were made up with acetonitrile or methanol at appropriate concentration levels. JZL-184, NAM, MAFP, THA and WWL70 were from Tocris Bioscience (Biogen, Madrid, Spain). R-flurbiprofen was from Sigma-Aldrich.

### 4.2. Animals

A total of 25 eight-week-old Sprague–Dawley rats were obtained from SGIker facilities (University of the Basque Country, UPV/EHU, Spain) and kept in a controlled environment (12 h light–dark cycle, 22 ± 2 °C and 55 ± 5% relative humidity) with food and water provided ad libitum for at least two weeks until they were sacrificed by decapitation under deep anesthesia (isoflurane 2–4%) at 10–12 weeks of age. Immediately after sacrifice, the rat brains were removed and then the cerebral cortices were dissected and stored at −80 °C until use.

Rats were handled following the guidelines of the Directive of the European Commission (2010/63/EU) and Spanish regulations (RD 53/2013) for the care and management of experimental animals, and the protocol of sacrifice was approved by the Committee of Ethics for Animal Welfare of the University of the Basque Country (UPV/EHU; CEBA/164/2010).

### 4.3. Tissue Sampling and Biochemical Fractionation for Enzymatic, Western Blot and Double Immunofluorescence Assays

Crude membrane (P2), cytosolic (S3) and microsomal (P3) fractions from brain cortices were obtained as previously described with minor variations [81,82]. Three pools of frozen cerebral cortices from three animals each were processed independently for fractionation. Brains were thawed and placed in pre-chilled lysis buffer (10 mM Tris, pH 7.6, 320 mM sucrose, 5 mM EDTA) before being homogenized mechanically with a motor-driven glass/glass tissue homogenizer in ten volumes of ice-cold lysis buffer containing protease inhibitors (1 mM phenylmethylsulfonyl fluoride -PMSF- and 0.5 mM iodoacetamide). One-tenth of the whole homogenate (WH) was aliquoted and stored at −80 °C until use, while the rest was centrifuged at low speed (1000× *g* for 10 min). The resulting supernatant was kept and the pellet (P1 fraction), containing cell nuclei and debris, was resuspended in the initial volume of lysis buffer and centrifuged again at 1000× *g* for 10 min. The final pellet was discarded and the S1 supernatants resulting from the two centrifugations were pooled and centrifuged at medium speed (10,000× *g*) for 15 min to separate a pellet of crude membranes (P2) and a cytoplasm supernatant (S2 fraction). The P2 pellet was resuspended in the initial volume of lysis buffer, divided in 1.5 mL Eppendorf tubes into equal volumes, and centrifuged again at medium speed. The supernatants were carefully aspirated from the pellets, pooled, and added to the initial S2 fraction, whereas the resulting washed final P2 fraction aliquots were stored at −80 °C until use. The cytoplasm supernatant (S2 fraction) was subjected to ultracentrifugation to separate a microsome pellet (P3) from cytosol (S3). The P3 pellet was resuspended in a small volume of lysis buffer and the P3 and S3 fractions were aliquoted into equal volumes and stored at −80 °C until use.

To isolate highly purified intact nuclei (N fraction) we followed the procedure described by Thompson and colleagues [83] with slight modifications [31,35,84]. Two pools of frozen cerebral cortices from eight adult rats each were processed independently. Tissue was thawed and chopped finely in 1 mM MgCl_2_, containing 2.0 M sucrose and protease inhibitors (1 mM PMSF and 0.5 mM iodoacetamide) and homogenized to give a 20% (w/v) homogenate. The homogenate was then filtered through one layer of muslin and centrifuged at 4 °C for 60 min at 64,000× *g* in a SW40Ti rotor (331302; Beckman). The resulting nuclear pellet was resuspended in 1 mM MgCl_2_ containing 320 mM sucrose and protease inhibitors, divided in 1.5 mL Eppendorf tubes into equal volumes, and centrifuged for 5 min at 1500× *g*. The supernatant was carefully aspirated from the pellets and the washed final N fraction aliquots were stored at −80 °C until use, except for nuclei used for immunofluorescence analysis, which were immediately resuspended in 10 mM Tris–HCl, pH 7.2 containing 2 mM MgCl_2_ at a density of 2 × 10^6^ nuclei/mL, laid on gelatin-coated slides (2 × 10^4^ nuclei/slide) and air dried before immunofluorescence staining.

Sequential fractionation of intact nuclei (N fraction) into nucleoids (Nu), nuclear matrix (NM), TX-100 extractable supernatant (S1) and DNase I/high salt extractable supernatant (S2) was carried out as described previously [85,86] with minor modifications [35]. Three-quarters of the N fraction obtained from the brain cortices of each eight-animal pool (~3.5 mg total protein/pool) were resuspended to 10 mg protein/mL in CSK 100 buffer (100 mM NaCl, 300 mM sucrose, 3 mM MgCl_2_, 0.5% TX-100, 0.5 mM CaCl_2_, 10 mM Pipes, 1.2 mM PMSF, 50 µM Iodoacetamide, pH 6.8) and subjected to 5 passages through a 21-gauge needle followed by incubation on ice for 7 min. The suspension was then divided in 1.5 mL Eppendorf tubes into equal volumes and subjected to centrifugation at 650× *g* for 5 min at 4 °C. After repeating this step, the supernatants resulting from the two centrifugations corresponding to the TX-100-solucle nuclear fraction (S1) were combined, divided into aliquots and stored at −80 °C. One-quarter of the nucleoid pellets (Nu), consisting of nuclei devoid of the envelope, was stored at −80 °C, whereas the rest (~2.5 mg) were resuspended together as above in 1250 µL CSK 50 buffer (50 mM NaCl, 300 mM sucrose, 3 mM MgCl_2_, 10 mM pipes, 1.2 mM PMSF, 50 µM iodoacetamide, pH 6.8) and incubated for 20 min at 37 °C with 100 IU (~40 IU/mg protein) recombinant RNase-free DNase I (2270A, Takara Bio Inc., Madrid, Spain). Subsequently, proteins were released from chromatin by adding, dropwise, 190 µL 2 M (NH_4_)2SO_4_ to achieve a final concentration of ~0.25 M followed by 15 min incubation on ice. The nuclear matrix (NM) was pelleted at 2000× *g* for 10 min. The pellet was then resuspended in 1440 µL CSK50 buffer, into equal volumes and subjected to an additional centrifugation at 2000× *g* for 10 min. The DNase I supernatants resulting from the two centrifugations corresponding to nuclear DNA-associated protein fraction (S2) were combined and aliquoted. Both, the NM pellet and S1 supernatant aliquots were stored at −80 °C until use.

Protein concentrations of all fractions were determined with the bicinchoninic acid (BCA) Protein Assay Kit (ab102536, Abcam, Madrid, Spain) using bovine γ-globulin as standard, except that of the DNase I supernatant, which was inferred from the loss of protein after treatment of nucleoids with DNase I and 0.25 M (NH_4_)_2_SO_4_ to obtain the NM fraction.

### 4.4. DGL Enzymatic Assays

To investigate the functionality of enzymes involved in the production and degradation processes of 2-AG, we measured the production in the nuclear matrix (NM) fractions derived from whole intact nuclei obtained from adult rat cerebral cortex, following the assay published previously (Shonesy et al., 2013 [36]) for the DGL enzymatic activity present in mouse striatal plasma membranes with some modifications. To avoid the nonspecific binding of reaction products and substrates, we performed the assays on 2 mL siliconized microcentrifuge tubes. 2-AG and AA accumulation were initiated by the addition of cold isolated nuclear matrix (10 μg nuclear protein, see the “Results” section) to the prewarmed assay buffer containing the appropriate concentrations of the selected substrates (SAG, 2-AG or 1-AG) or inhibitors of DGLα, MGL, ABHD6, ABHD12 and FAAH in a final volume of 50 μL. The assay buffer contained 20 mM HEPES, pH 7.5, and 10 mM DTT. These assays were run for 30 min at 37 °C in gentle agitation and were stopped by keeping on ice. Tubes were spiked with 2.5 μL acetonitrile containing the internal standards (final concentrations: 2-AG-d5 520 nM, and AA-d8 2 µM). Ethyl acetate/hexane (1000 μL; 9:1, v/v) was added to extract the nuclear fraction with the aid of a 5 mm-steel ball using the TissueLyser II (Qiagen, Hilden, Germany) for 1 cycle of 10 s at 30 Hz. Then the tubes were centrifuged for 10 min at 10,000× *g* and 4 °C, and the upper (organic) phase was removed, evaporated to dryness under a gentle stream of nitrogen at 20–25 °C temperature and re-dissolved in 50 μL acetonitrile. After sonication and rape again, the analysis was performed as previously described [35,87] on a LC-MS/MS system based on Agilent technologies (Wilmington, DE, USA) consisting of a 6410 Triple Quad mass spectrometer equipped with an electrospray ionization source operating in positive ion mode, and a 1200-series binary pump system. Natural and deuterated 2-AG and AA were separated with a Phenomenex Luna 2.5 μm C18(2)-HST column, 100 × 2 mm, combined with a Security Guard pre-column (C18, 4 × 2 mm; Phenomenex) with solvents A (0.1% formic acid in 20:80 acetonitrile/water, v/v) and B (0.1% formic acid in acetonitrile), using the following gradient: 55–90% B (0–2 min), then held at 90% B (2–7.5 min) and re-equilibrated at 55% B (7.5–10 min). The column temperature was 25 °C, the flow rate was 0.3 mL/min, the injection volume was 10 μL and the needle was rinsed for 60 s using a flushport with water/acetonitrile (80:20) as the eluent. The electrospray ionization interface was operated using nitrogen as a nebulizer and desolvation gas, and using the following settings: temperature 350 °C, nebulizer pressure 40 psi, and capillary voltage +4800 V. The following precursor-to-product ion transitions were used for multiple-reaction monitoring (MRM): 2-AG m/z 379→287; 2-AG-d5 m/z 384.2→287; AA m/z 305→91; AA-d8 313→126. Dwell times were 20 ms; the pause between MRM transitions was 5 ms. Data acquisition and analysis were performed using MassHunter Software.

### 4.5. Western Blot Assays

Briefly, known amounts of total protein from the different samples were heat-denatured for 5 min in urea-denaturing buffer (20 mM Tris-HCl, pH 8.0, 12% glycerol, 12% urea, 5% DTT, 2% SDS, 0.01% bromophenol blue). Proteins were resolved by electrophoresis in 10% SDS-PAGE gels in a Mini Protean II gel apparatus (Bio-Rad; Hercules, CA, USA) followed by Coomassie blue dye staining or transference to polyvinylidene fluoride (PVDF) membranes (Amersham Biosciences, Piscataway, NJ, USA) at 30 V overnight at 4 °C with a Mini TransBlot transfer unit (Bio-Rad; Hercules, CA, USA). For immunoblot, PVDF Blots were blocked at 20–25 °C for 1 h in a blocking solution consisting of 0.2 M phosphate-buffered saline pH 7.4 (PBS), containing 5% non-fat dry milk (1706404, Bio-Rad), 0.5% bovine serum albumin (BSA, Sigma-Aldrich), 0.2% Tween-20 (Sigma-Aldrich), followed by overnight incubation at 4 °C with primary antibodies in blocking solution without milk (Table 1 for details). After three washes (10 min each) at 20–25 °C with PBS containing 0.1% Tween-20, blots were incubated for 2 h at 20–25 °C with horseradish peroxidase (HRP)-conjugated rabbit anti-goat IgG (A5420; Sigma-Aldrich), HRP-conjugated donkey anti-rabbit IgG (NA934; Amersham Biosciences) or HRP-conjugated sheep anti-mouse IgG (NXA931; Amersham Biosciences) secondary antibodies, all diluted to 1:10,000 in blocking solution. Immunoreactive bands were visualized by enhanced chemiluminescence using the Clarity Western ECL Substrate (#1705061; Bio-Rad Laboratories) according to the manufacturer’s instructions.

### 4.6. Double Immunofluorescence

Nuclei were laid on gelatin-coated slides and were immersed in buffered 2% paraformaldehyde for 15 min at 20–25 °C and processed for double immunofluorescence. Briefly, nuclei were washed extensively with wash buffer (PBS containing 0.22% gelatin) and incubated in blocking solution (wash buffer containing 1% serum albumin bovine -BSA; Sigma-Aldrich- and 1% normal donkey serum, Jackson ImmunoResearch Laboratories Inc.; West Grove, PA, USA-) for 1 h at 20–25 °C, followed by overnight incubation at 4 °C with the corresponding combination of primary antibodies (Table 1). After three washes (10 min each) at 20–25 °C with washing buffer, nuclei were incubated for 1 h at 20–25 °C with a combination of Alexa Fluor 488 donkey anti-goat IgG (A11055, Invitrogen) and DyLight 549 donkey anti-rabbit F(ab’)2 fragment (711-506-152, Jackson ImmunoResearch) diluted 1:400 in blocking solution, extensively washed and counterstained with Hoechst’s chromatin staining counterstained with 0.1 μg/mL Hoechst 33,342 (Sigma-Aldrich) in wash buffer, for 10 min at 20–25 °C. After two additional washes at 20–25 °C with PBS, slides were coverslipped with homemade Mowiol (Calbiochem, Madrid, Spain) mounting medium containing anti-fade reagent 1,4-phenylene-diamine dihydrochloride (Sigma-Aldrich).

### 4.7. Microscope Studies and Imaging

Doubly-labeled nuclei were observed with an epifluorescence microscope Carl Zeiss Axio Observer.Z1, equipped with an HXP 120 C metal halide lamp illumination source (Carl Zeiss MicroImaging, Inc., Gottigen, Germany) and digitally captured with a monochromatic camera (AxioCam MRm, 1388 × 1040 pixels). Images were taken using a 63× Plan-Apochromat objective (NA 1.4), and optical sections (0.24 µm intervals in the z-axis) were obtained with the Zeiss ApoTome structured illumination module using a XYZ motorized stage from Carl Zeiss MicroImaging, Inc. Camera settings adjusted to obtain images with a pixel size of 0.01 μm^2^. Bandpass filters were 49 DAPI (Ex G 365, Em 445/50) for Hoecsht’s staining, 38 HE eGFP (Ex 470/40, Em 525/50) for Alexa Fluor 488, and 43 HE Cy3 shift-free (Ex 550/25, Em 605/70) for DyLight 549. Images were digitized using Zeiss Axio Vision 4.8 software, exported to TIFF format, and compiled and labeled using Adobe Photoshop CS3 after minor despeckling on Fiji-ImageJ 1.53f51 (NIH, Bethesda, MA, USA).

### 4.8. Data Analysis

GraphPad Prism (version 5.0, GraphPad Software Inc., San Diego, CA, USA) was used to organize and statistically analyze data. The concentration-response curves for MAFP were analyzed by nonlinear regression using the appropriate equations, which gave estimates of the basal level, maximal response (*E_max_*) and *IC_50_* of the curves. The data are usually presented as mean ± SEM for the indicated number of experiments, carried out by using separate nuclear preparations performed in duplicate. The significance of differences between means was analyzed by unpaired two-tailed Student’s *t*-test or one-way analysis of variance (ANOVA) followed by Tukey–Kramer multiple comparison tests. Statistical significance was set at the 95% confidence level.

## Figures and Tables

**Figure 1 ijms-24-03165-f001:**
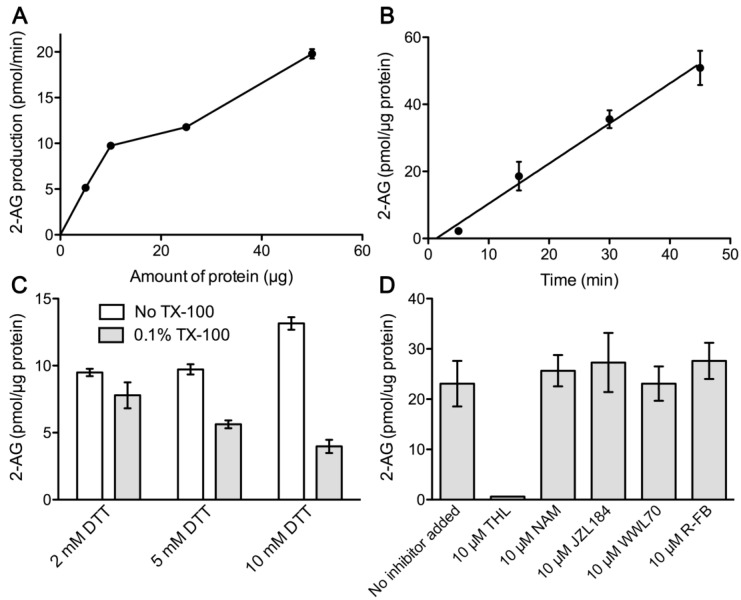
Optimization assays for the measurement of DGL activity-dependent 2-AG production in samples of the nuclear matrix obtained from intact nuclei of the adult rat cortex using 200 µM SAG as substrate. (**A**) Analysis of 2-AG production as the amount of nuclear matrix protein increases. (**B**) Analysis of 2-AG accumulation in the nuclear matrix over time. (**C**) Influence of the reducing agent DTT and of the detergent TX-100. (**D**) Effects of DGL, serine-hydrolase and COX2 inhibitors. Incubation of samples with the nonselective DGL inhibitor tetrahydrolipstatin (THL) completely abrogated 2-AG accumulation. MGL inhibitors, N-arachidonoyl maleimide (NAM) and JZL184; ABHD6 inhibitor, WWL70; COX2 inhibitor, R-flurbiprofen, (R-FB). Data are mean ± SEM (n = 3).

**Figure 2 ijms-24-03165-f002:**
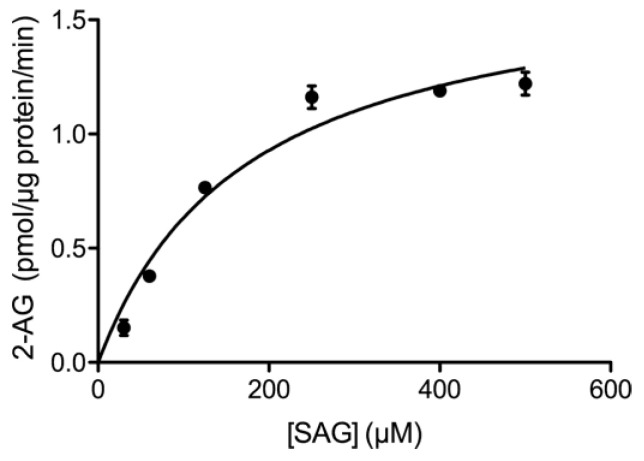
Analysis of the enzyme kinetics of DGL activity in nuclear matrix samples obtained from intact nuclei of the adult rat cortex. Increasing amounts of the DGL substrate SAG were added and 2-AG was measured by LC/MS/MS. Data are mean ± SEM (n = 3).

**Figure 3 ijms-24-03165-f003:**
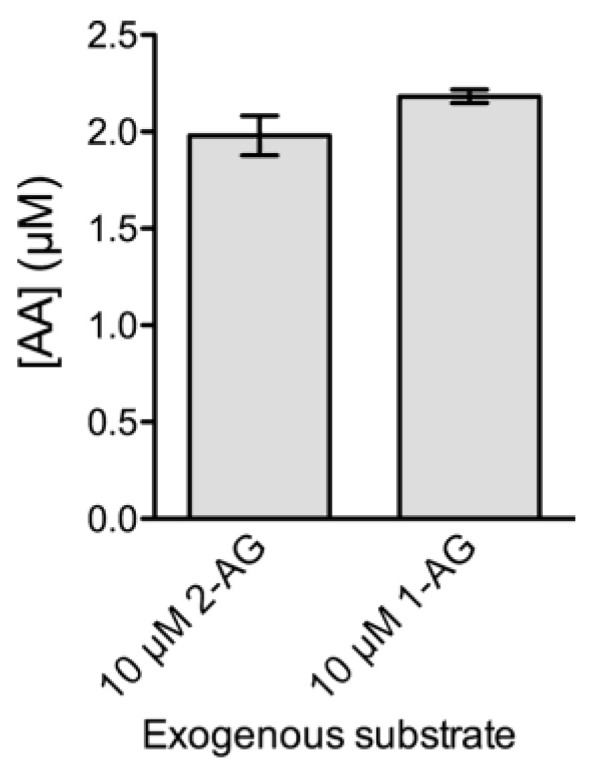
Accumulation of arachidonic acid (AA) during 30 min in 10 µg protein-containing nuclear matrix samples purified from intact cortical nuclei. 2-AG or 1-AG (both at 10 µM) were used as exogenous substrates. Data are mean ± SEM (n = 3).

**Figure 4 ijms-24-03165-f004:**
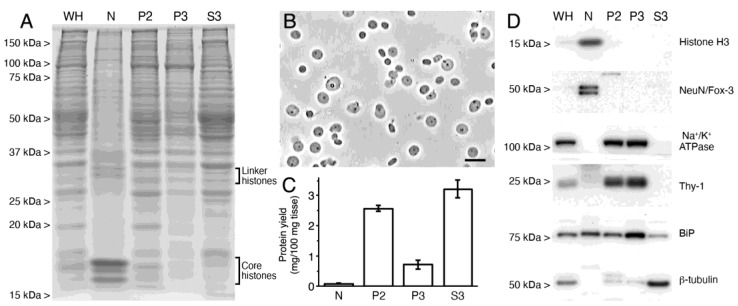
Characterization of subcellular fractions obtained from adult rat brain cortex. (**A**) Coomassie blue-stained SDS-PAGE of equivalent amounts of protein from whole homogenate (WH) and subcellular fractions: intact nuclei (N), plasma membrane (P2), microsomes (P3) and cytosol (S3). (**B**) Representative image of phase-contrast microscopy of nuclei isolated from a homogenate of adult rat cerebral cortex. Scale bar = 20 µm. (**C**) Bar graph depicting the partitioning of total protein from rat cortical homogenate into N, P2, P3 and S3 fractions expressed as mg of protein recovered in each fraction per 100 mg of rat cortical tissue. Data represent the mean ± SEM of the values obtained in three (P2, P3 and S3 fractions; n = 3) or two (N fraction; n = 2) independent fractionations. (**D**) Western blot analysis of subcellular fractions obtained from homogenates of rat brain cortex run in parallel to analyze the suitability of the subfractioning procedure using specific markers for nuclear (Histone H3 and NeuN/Fox-3), plasma membrane (Na^+^/K^+^ ATPase and Thy-1), endoplasmic reticular (BiP) and cytosolic (β-tubulin) fractions.

**Figure 5 ijms-24-03165-f005:**
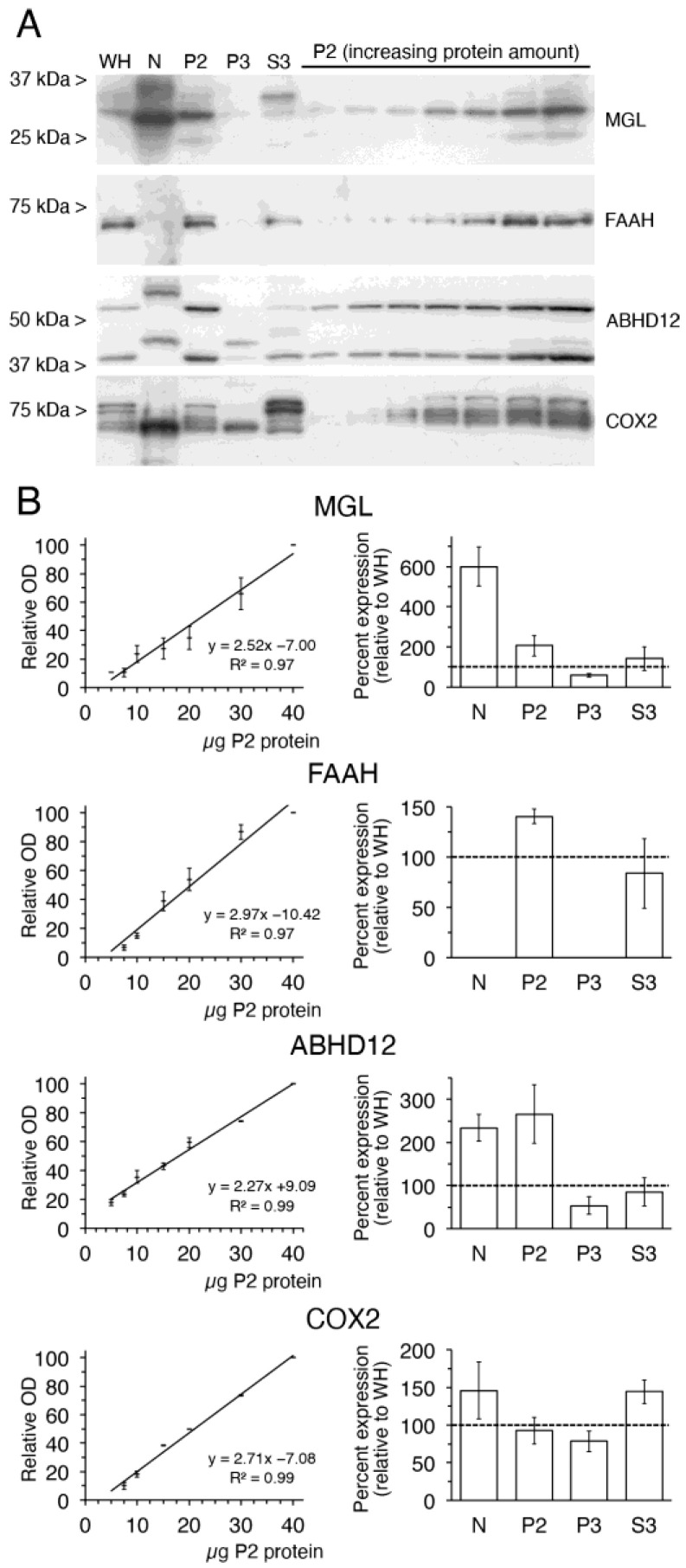
Analysis of the subcellular expression of 2-AG metabolizing enzymes monoacylglycerol lipase (MGL), fatty acid amide hydrolase (FAAH), α/β hydrolase domain-containing protein 12 (ABHD12) and cyclooxygenase-2 (COX2). (**A**) Representative immunoblots of whole homogenate (WH) intact nuclear (N), plasma membrane (P2), microsomal (P3) and cytosolic (S3) fractions obtained from homogenates of the adult rat brain cortex. 20 µg protein from each fraction were loaded in the same gel. Incremental amounts of P2 membranes ranging from 5 to 40 µg protein were also run in parallel to generate reference standard curves used for semi-quantitative densitometric analysis of protein expression. (**B**) Graphs on the left correspond to standard curves of MGL-, FAAH-, ABHD12- and COX2-immunoreactivity in P2 samples. All optical density (OD) values were normalized to the value obtained at the highest protein loading. The correlation coefficients obtained by linear regression analysis are shown. Bar graphs on the right show results of the semi-quantitative analysis of MGL, FAAH, ABHD12 and COX2 protein expression in N, P2, P3 and S3 fractions. Values correspond to the percentage of the expression found in WH (100%, dashed line). Data presented are means ± SEM of four independent experiments (n = 4), except for FAAH (n = 2).

**Figure 6 ijms-24-03165-f006:**
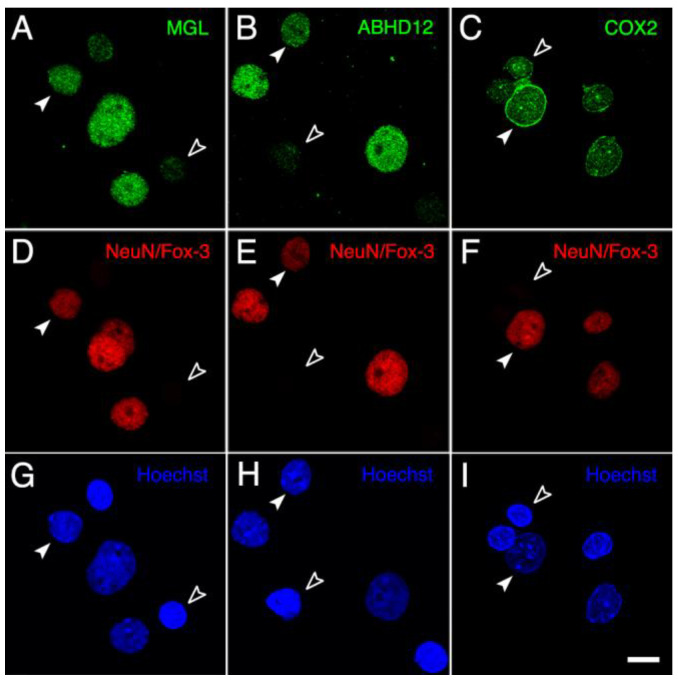
Panoramic micrographs of intact nuclei isolated from the adult rat brain cortex processed for MGL-, ABHD12- and COX2-immunofluorescence (**A**–**C**) combined with NeuN/Fox-3-immunofluorescence (**D**–**F**) and Hoechst’s chromatin staining (**G**–**I**). Every NeuN-positive nucleus exhibited strong immunoreactivity for MGL, ABHD12 and COX2 (filled arrowheads), whereas NeuN-negative nuclei (empty arrowheads) displayed clear positive staining only for COX2. All micrographs were acquired in grayscale and pseudo colored. Scale bar = 20 µm in I (applies to (**A**–**I**)).

**Figure 7 ijms-24-03165-f007:**
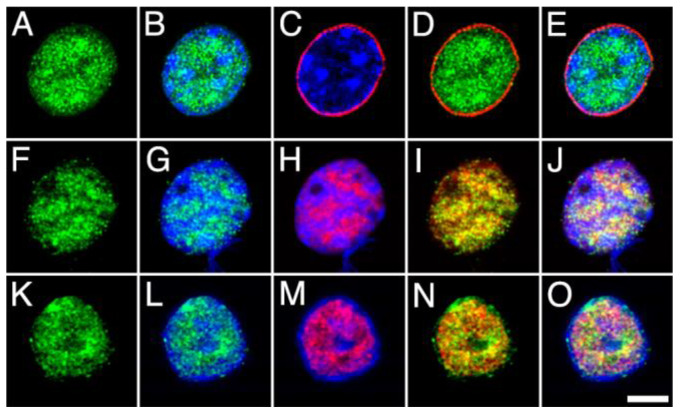
High-power micrographs of intact nuclei isolated from the adult rat brain cortex double stained with anti-MGL (green) combined with Hoechst staining (blue) and NPCx ((**A**–**E**); red), NeuN/Fox-3 ((**F**–**J**); red) and DGLα ((**K**–**O**); red) immunostaining. All micrographs are maximum intensity projections of three consecutive 0.24 µm-thick optical sections acquired in grayscale and pseudo colored. Scale bar = 5 µm in O (applies to (**A**–**O**)).

**Figure 8 ijms-24-03165-f008:**
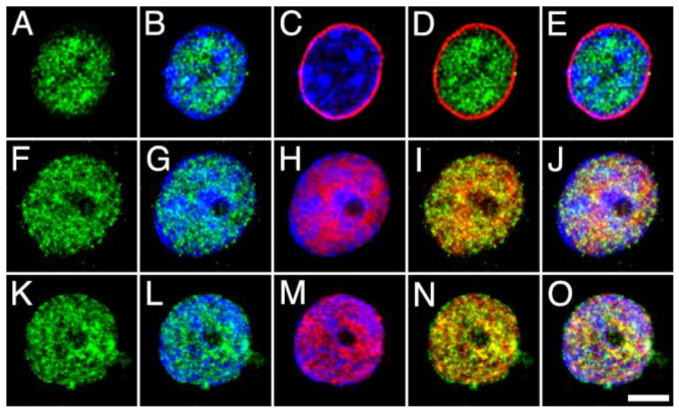
High-power micrographs of intact nuclei isolated from the adult rat brain cortex double stained with anti-ABHD12 (green) combined with Hoechst staining (blue) and NPCx ((**A**–**E**); red), NeuN/Fox-3 ((**F**–**J**); red) and DGLα ((**K**–**O**); red) immunostaining. All micrographs are maximum intensity projections of three consecutive 0.24 µm-thick optical sections acquired in grayscale and pseudo colored. Scale bar = 5 µm in O (applies to (**A**–**O**)).

**Figure 9 ijms-24-03165-f009:**
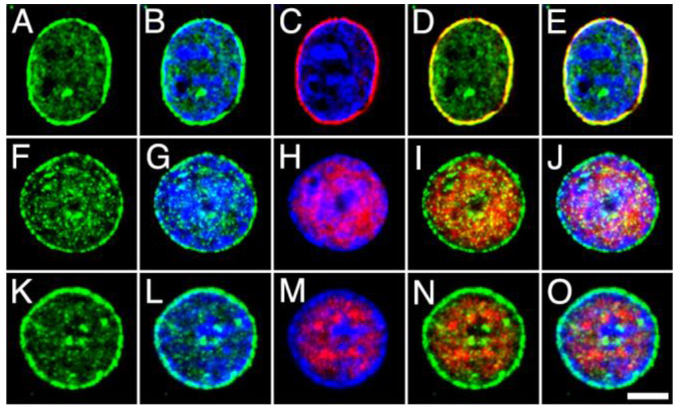
High-power micrographs of intact nuclei isolated from the adult rat brain cortex double stained with anti-COX2 (green) combined with Hoechst staining (blue) and NPCx ((**A**–**E**); red), NeuN/Fox-3 ((**F**–**J**); red) and DGLα ((**K**–**O**); red) immunostaining. All micrographs are maximum intensity projections of three consecutive 0.24 µm-thick optical sections acquired in grayscale and pseudo colored. Scale bar: 5 µm in O (applies to (**A**–**O**)).

**Figure 10 ijms-24-03165-f010:**
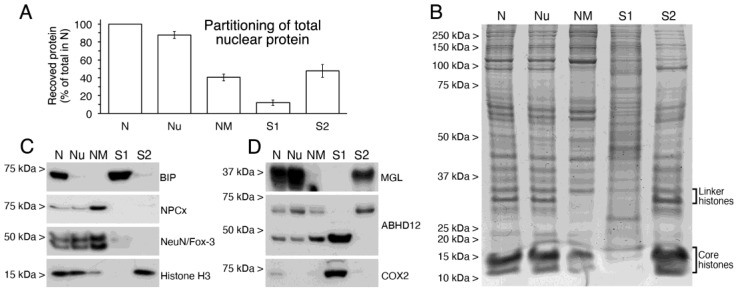
Characterization of nuclear subfractions obtained from intact nuclei and analysis of nuclear compartmentation of MGL, ABHD12, COX2. (**A**) Bar graph depicting the partitioning of total protein from cortical intact nuclei (N) into nucleoids (Nu), nuclear matrix (NM), TX-100 extractable supernatant (S1) and DNase I/high salt extractable supernatant (S2). Data are expressed as percent protein recovered in each fraction relative to total protein (100%) in N fraction. Values are mean ± SD of two independent experiments. (**B**) Coomassie blue-stained SDS-PAGE of equivalent amounts (20 µg) of protein from N, Nu, NM, S1 and S2 samples. (**C**) Western blot analysis of nuclear subfractions (12 µg/lane) run in parallel to analyze the suitability of the subfractioning procedure using specific markers for non-ionic detergent extractable (BiP), non-ionic detergent and DNase I/high salt resistant (NPCx), nuclear matrix (NeuN/Fox-3) and DNA-bound (Histone H3) proteins. (**D**) MGL, ABHD12 and COX2 immunoblots in N, Nu, NM, S1 and S2 nuclear subfractions. 12 µg/lane were loaded.

**Figure 11 ijms-24-03165-f011:**
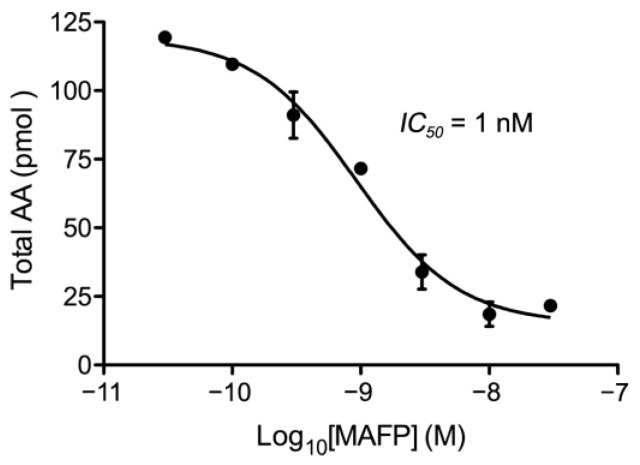
Inhibition curve of AA accumulation in samples of nuclear matrix obtained from intact nuclei of the adult rat cortex using increasing concentrations (0.03–30 nM) of the nonselective serine-hydrolase inhibitor MAFP. Data are mean ± SD (n = 2).

**Figure 12 ijms-24-03165-f012:**
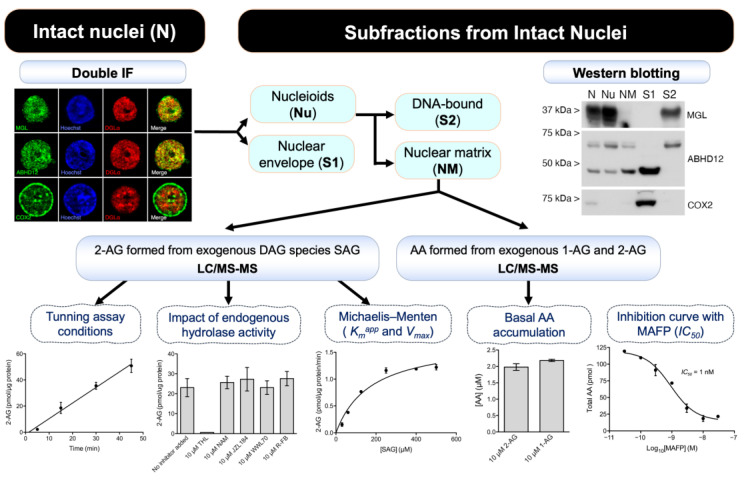
Flow diagram highlighting the key experiments and results leading to the conclusion that ABHD12 is likely the main responsible for the hydrolysis of 2-AG in the nuclear matrix.

**Table 1 ijms-24-03165-t001:** Primary antibodies used.

Target	Dilution		Host/Clonality	Isotype	Antigen	Reference
	IF	WB				
ABHD12	1:200	1:2500	Goat polyclonal	Affinity purified (ns)	C-REFLGKSEPEHQH peptide	Sigma-Aldrich, SAB2500016
COX2	1:200	1:2500	Goat polyclonal	IgG	Peptide located at the C-terminus of human COX2	S^ta^ Cruz Biotech., sc-1745
DGLα	1:250	1:1000	Rabbit polyclonal	Affinity purified (ns)	Peptide corresponding to the 42 amino acids of the C-termninal end of mouse DGLα	Frontier Science, DGLα-Rb-Af380-1
FAAH	−	1:1000	Goat polyclonal	IgG	Peptide mapping near the N-terminus of human FAAH	S^ta^ Cruz Biotech., sc-26427
GRP78/BiP	−	1:2000	Rabbit polyclonal	IgG	Peptide corresponding the 55 C-terminal aas of GRP78/BiP	Abcam, ab21685
Histone H3	−	1:500	Rabbit polyclonal	IgG	Peptide corresponding to aas 1–20 of histone H3	Chemicon, 382157
MGL	1:200	1:1000	Goat polyclonal	IgG	Peptide corresponding to aas 17–29 of human MGL	Abcam, ab77398
Na^+^/K^+^ ATPase	−	1:5000	Mouse monoclonal	IgG_1_	α1 subunit of Na+/K+ -ATPase purified from lamb liver	Sigma-Aldrich, A-277
NeuN/Fox-3	1:1000	1:2000	Mouse monoclonal	IgG_1_	Nuclei purified from mouse brain	Chemicon, MAB377
NPCx	1:4000	1:5000	Mouse monoclonal	IgG_1_	Protein mixture of the nuclear pore complexes	Abcam, ab24609
Thy1/CD90	−	1:6000	Rabbit monoclonal	IgG	Epitope located near the N-terminus of human Thy1/CD90	Abcam, ab92574
β-tubulin	−	1:1000	Mouse monoclonal	IgG_1_	Epitope between aas 281–446 at the C-terminus of β-tubulin	Sigma-Aldrich, T4026,

ns, not specified by the vendor.

## Data Availability

The data presented in this study are available within the article. Additional inquiries may be directed to the corresponding authors.

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
