# Peer review of "Endocannabinoid 2-Arachidonoylglycerol Synthesis and Metabolism at Neuronal Nuclear Matrix Fractions Derived from Adult Rat Brain Cortex"

_ijms, 2023, doi:10.3390/ijms24043165_

Round 1
Reviewer 1 Report
The paper in my opinion is accettable in present form
Author Response
We are very grateful to the Reviewer for his/her very kind words.
Reviewer 2 Report
The authors here describe how diacylglycerol lipase-α (DGLα) located at the nuclear matrix plays a crucial role in the determining the kinetics and turnover of 2-AG. The authors use various biochemical and imaging techniques to show and pinpoint these mechanisms. This is very interesting work which will not only help basic science researchers, but also help pharmacologists top determine better strategies for designing drugs which are specifically targeted towards ABDH12 located in the neuronal matrix. Although the study designs look good there are a few minor things that need to be added to make the findings even more solid.
1. The authors need to highlight in depth either in the introduction or discussion section the commercial and pharmacological importance of this study.
2. In the section 2 the authors have used some inhibitors which mostly target ABDH, MGL and COX. But the authors didn’t use any FAAH inhibitors, would be good if the authors could either show a small study with FAAH inhibitors like URB571 to see if there any effects on 2 AG hydrolysis, as certain studies have pointed out the role of FAAH in this (PMID: 18394720)
3. Additionally, the authors could also carry out some activity-based proteomics -using MAFP to begin with, coupled with current studies to see if there are any other enzymes (novel or know) which might be having any effect in this phenomenon.
4. It will be also helpful for the reader if the authors provide a figure/ flowchart highlighting the key experimental procedures and the main results.
Author Response
Comments and Suggestions for Authors
The authors here describe how diacylglycerol lipase-α (DGLα) located at the nuclear matrix plays a crucial role in the determining the kinetics and turnover of 2-AG. The authors use various biochemical and imaging techniques to show and pinpoint these mechanisms. This is very interesting work which will not only help basic science researchers, but also help pharmacologists top determine better strategies for designing drugs which are specifically targeted towards ABDH12 located in the neuronal matrix. Although the study designs look good there are a few minor things that need to be added to make the findings even more solid.
We sincerely thank this reviewer for his comments to improve the manuscript and, most importantly, for the really interesting and attractive ideas that he/she proposes and that encourage us to continue in this line of research.
Please find below a response to the reviewer's specific comments and suggestions.
1. The authors need to highlight in depth either in the introduction or discussion section the commercial and pharmacological importance of this study.
Thank you very much for bringing this interesting point to our attention. We have included the following sentence in the discussion that highlights the potential ABHD12 as a drug target:
“Although these hypotheses are still very premature and require further investigation, our results and the body of evidence in the literature suggest that the noncanonical pathway for 2-AG production in the neuronal nucleus may have relevant pathophysiological implications. Therefore, the importance of this signaling pathway for drug development deserves attention. Although clinical research has focused on potent inhibitors of FAAH and MAGL, with drugs even being licensed for clinical trials or neuroinflammatory and neurological research [79,80], the development of compounds that selectively inhibit enzymes of the ABHD family, including ABHD12, is still in its infancy. Our results, which show that ABHD12 is probably the most important serine hydrolase that tunes 2-AG production in the nuclear matrix, point to this enzyme as a potential drug target. Furthermore, ABHD12 inhibitors should be evaluated with this framework in mind.”
2. In the section 2 the authors have used some inhibitors which mostly target ABDH, MGL and COX. But the authors didn’t use any FAAH inhibitors, would be good if the authors could either show a small study with FAAH inhibitors like URB571 to see if there any effects on 2 AG hydrolysis, as certain studies have pointed out the role of FAAH in this (PMID: 1839472.0)
Thank you very much for this comment. As noted by the reviewer, and described both in the review that she/he mentions (now included as ref. 28) and in the experimental data (ref. 54), the redundancy of 2-AG hydrolyzing enzymes suggests caution and recommends the use of selective inhibitors to rule out any impact on FAAH on 2-AG accumulation. However, our data conclusively show the absence of a detectable signal for this enzyme in isolated nuclei from the rat cerebral cortex, which led us to discard this possibility.
3. Additionally, the authors could also carry out some activity-based proteomics -using MAFP to begin with, coupled with current studies to see if there are any other enzymes (novel or know) which might be having any effect in this phenomenon.
We are very grateful for this very interesting suggestion that we would like to address in the future on our laboratory bench to get further insight on the protein players involved in 2-AG-mediated signaling. We have mentioned this attractive strategy to contribute to the knowledge and search for pharmacological targets.
“As addressed in mouse brain membranes, an activity-based protein profiling (ABPP) approach [25] in combination with nonselective inhibitors and serine hydrolases could shed light on the contribution of different molecular entities modulating 2-AG levels in different nuclear compartments.”
4. It will be also helpful for the reader if the authors provide a figure/ flowchart highlighting the key experimental procedures and the main results.
Thank you very much again for that comment. We have generated Figure 12 that we believe facilitates the understanding of what are the key assays and the workflow that lead to the main results.
Reviewer 3 Report
The authors provide that subcellular distribution of neuronal DGL, and the 2-AG is produced in the neuronal nuclear matrix by carrying out biochemical and morphological evidence. This is very interesting study in endogenous cannabinods system. Given that CB1 and CB1 are very important drug targets for drug discovery, and recent studies has deciphered more mechanism of CB1 receptor (DOI:10.1038/s41589-022-01038-y ), could the author provide more information of relationship of 2-AG location with disease treatments? Since some 2-ag was reported to be involved in pain relief and some ligands of CB1 participates in pain management as well.
Author Response
This is very interesting study in endogenous cannabinods system. Given that CB1 and CB1 are very important drug targets for drug discovery, and recent studies has deciphered more mechanism of CB1 receptor (DOI:10.1038/s41589-022-01038-y), could the author provide more information of relationship of 2-AG location with disease treatments? Since some 2-AG was reported to be involved in pain relief and some ligands of CB1 participates in pain management as well.
We are very grateful to the reviewer for his interesting comments on the possibility that nuclear 2-AG could be involved in pain relief, and we have included the following paragraph in the discussion section, with two new references:
“Another potential target of 2-AG belonging to the same family of transcription factors as PPARγ is the related nuclear receptor PPARα [75], whose activation is known to has anti-inflamatory effects [76]. Interestingly, pharmacological blockade of endocannabinoid-inactivating hydrolases reverses PPARα-mediated anti-nociceptive effects [77]. In this regard, it has recently been proposed that AEA exerts part of its anti-nonciceptive effects through PPARα [78]. In view of our results, locally produced 2-AG in the nuclear matrix and its hydrolyzing enzymes could be equally candidates for a role in PPARα-dependent molecular mechanisms of pain.”
Round 2
Reviewer 3 Report
I have no further question for this manuscript